# Scalable production of ultrafine polyaniline fibres for tactile organic electrochemical transistors

Bo Fang [1,2,7 ✉], Jianmin Yan[1,3,7], Dan Chang[4], Jinli Piao[1,2], Kit Ming Ma[1,2], Qiao Gu[5,6], Ping Gao[5,6], Yang Chai [1,3 ✉] & Xiaoming Tao [1,2 ✉]

The development of continuous conducting polymer fibres is essential for applications ranging from advanced fibrous devices to frontier fabric electronics. The use of continuous conducting polymer fibres requires a small diameter to maximize their electroactive surface, microstructural orientation, and mechanical strength. However, regularly used wet spinning techniques have rarely achieved this goal due primarily to the insufficient slenderization of rapidly solidified conducting polymer molecules in poor solvents. Here we report a good solvent exchange strategy to wet spin the ultrafine polyaniline fibres. The slow diffusion between good solvents distinctly decreases the viscosity of protofibers, which undergo an impressive drawing ratio. The continuously collected polyaniline fibres have a previously unattained diameter below 5 μm, high energy and charge storage capacities, and favorable mechanical performance. We demonstrated an ultrathin all-solid organic electrochemical transistor based on ultrafine polyaniline fibres, which operated as a tactile sensor detecting pressure and friction forces at different levels.

---

[1] Research Institute for Intelligent Wearable Systems, The Hong Kong Polytechnic University, 999077 Hong Kong, China. [2] Institute of Textiles and Clothing, The Hong Kong Polytechnic University, 999077 Hong Kong, China. [3] Department of Applied Physics, The Hong Kong Polytechnic University, 999077 Hong Kong, China. [4] Department of Polymer Science and Engineering, Zhejiang University, 310027 Hangzhou, China. [5] Department of Chemical and Biological Engineering, The Hong Kong University of Science and Technology, 999077 Hong Kong, China. [6] Advanced Materials Thrust, The Hong Kong University of Science and Technology (Guangzhou), 510000 Guangzhou, China. [7] These authors contributed equally: Bo Fang, Jianmin Yan. ✉email: bofang@polyu.edu.hk; ychai@polyu.edu.hk; xiao-ming.tao@polyu.edu.hk

The extended conjugated and easily doped π-system along the backbone enables conducting polymers to possess intriguing transport, optical, and electrochemical properties, which have rarely been found in conventional polymers and metal conductors[1–3]. Processing conducting polymers into macroscopically fibrous materials makes it possible to translate their nano-object features to human-friendly products in a continuous manner. The combined merits, including but not limited to mechanical flexibility, intrinsic conductivity, and electrochemical activity, of conducting polymer fibers (CPFs) have introduced an era of "electronic textiles"[4]. For instance, highly conductive and electrochemically active poly(3-methylthiophene) fibers have been achieved by in situ electrochemical oxidation of monomers[5]. Fast ion transport between CPFs and ionic liquids has given birth to long-term operation actuators, electrochromic windows, and numeric displays[6]. In recent studies, the wet-spun poly (3,4-ethylene dioxythiophene) (PEDOT) fibers have been widely used in various frontier fields, such as flexible energy storage electrodes, implantable bioelectronics, and organic transistors[7,8].

Unfortunately, due primarily to the large diameters, the performance and expectations of most achieved continuous CPFs have been limited by their insufficient electroactive surfaces and weak breaking strengths. Electrospinning and wet spinning are two mainstream strategies to produce continuous CPFs. In the case of electrospinning, the fairly rigid backbone due to the high aromaticity results in an insufficient elasticity of conducting polymer solutions, which fails to be solely electrospun into fine fibers[9]. Although a two-fluid electrospinning technique has been proposed by coating a soluble and electrospinnable fluid on the conducting polymer cores, the complex procedures involving the addition and removal of second components defy the mass production of electrospun CPFs[10,11]. In the case of conventional wet spinning, conducting polymer dopes tend to occur a transient solidification in poor solvents, induced by the strong interactions of conducting polymer chains. The rapidly hardened gels suppress the post-stretching and slenderizing procedures, and cause the wet-spun CPFs to show a large diameter, generally beyond 10 μm[12–14]. The large diameters largely discount the mechanical properties and electrochemical activities of CPFs[4,15]. Thus, there is an urgent need to realize the mass production of ultrafine CPFs, which remains challenging. In this work, we report a good solvent exchange strategy in a modified wet spinning technique to prepare the ultrafine polyaniline (PAni) fibers (UFPFs) at a large scale. Beyond conventional wet spinning protocol, we replaced poor solvents by good solvents as the coagulation bath to decrease the viscosity of gel protofibres, which were subject to an ultrahigh drawing ratio and reduced to an ultrafine morphology. The obtained UFPFs own a small diameter below 5 μm, an unprecedented mechanical strength of $1080 \pm 71$ MPa, a high area capacitance beyond $1008$ mF cm$^{-2}$, and an enormous charge storage capacity of $5.25 \times 10^4$ mC cm$^{-2}$. Based on the structural and electrochemical merits of UFPFs, we demonstrated an ultrathin all-solid organic electrochemical transistor (OECT) with less than 1 V driving voltage, which substantially amplified drain-source electrical signals with low power consumption and responded to vertical pressure and horizontal friction forces at different levels. Our work opens an avenue to prepare continuous ultrafine CPFs and high-performance soft electronics.

## Results

**Preparation and characterization of UFPFs.** In the modified one-step wet spinning process, we used good solvents as the coagulation bath to realize the mass production of UFPFs (Fig. 1a, b and Supplementary Fig. 1). After doping PAni powder (emeraldine base) with camphor sulfonic acid (CSA) at a molar ratio of 2:1, we dispersed fully doped PAni into m-cresol as the raw spinning dopes (see the Methods section)[16]. Significantly, the direct use of doped PAni solutions as the dopes saves the trouble of conventional post-doping procedures, and further permits the uniform charge distribution throughout the fiber length[17]. A good solvent, dimethyl formamide (DMF), of PAni operated as the coagulation bath. A slow solvent exchange between m-cresol and DMF facilitated the formation of PAni gel protofibres with a quite low viscosity below 3000 cP. Subsequently, we observed a sharp decrease of diameter from ~0.1 mm to ~4.7 μm when stretching the gel fibers in the bath (Fig. 1c–f), which, to our knowledge, is a record small value in the achieved wet-spun CPFs[4]. The ultrafine fiber shows a smooth surface (Fig. 1f and Supplementary Fig. 2), highly crystallized microstructures (Supplementary Fig. 3), and uniform electrical properties (Supplementary Fig. 4). Moreover, such an impressive drawing ratio enables a very high production efficiency of UFPFs beyond 40 m/min. For example, we prepared a 5.4-kilometers-long UFPF in 2 h (Fig. 1g).

The sharp necking behaviors of gel protofibres are highly related to the use of good solvents as coagulation bath. In a good solvent, interactions between PAni chains and solvent molecules are energetically favorable, and will cause PAni chains to expand and disperse well. In a poor solvent, the chain interactions are preferred and suppress the dispersion of PAni molecules. We recorded the evolution of surface morphologies of PAni fibers collected from different solvating species. As shown in Fig. 2a, the obtained fibers in poor solvating species, i.e., water, ethanol, ethyl acetate (EA), and acetone, generally present coarse surfaces and large diameters around 20 μm. By comparison, we clearly observed a necking phenomenon in both cases of good solvents, i.e., N-methyl-2-pyrrolidone (NMP) and DMF. Such necking effects promoted the finally produced fibers to behave ultrafine morphologies, which assists PAni fibers to behave a higher degree of orientation and crystallization (see the X-ray diffraction analysis in Supplementary Fig. 3). Further, the higher degree of crystallization keeps the doping bonding in PAni chains from the attack of ambientes at the molecular level, and is conducive to a better structure and performance stability. We used Raman spectra to evaluate their structural evolution after placing fibers in air for four weeks. As shown in Fig. 2b, we did not find obvious de-doping signals in Raman spectra of the PAni fibers from good solvents, whereas various de-doping peaks (1223 and 1462 cm$^{-1}$) appeared in the cases of poor solvents, which inflects the dissociation of proton-polymer interactions[18–20].

We speculate that this sharp necking phenomenon may be caused by two factors: diffusion difference and interfacial pressure. In the conventional wet spinning protocol, the diffusion from good solvents to poor solvents occurs quickly to solidify dope fluids into gel fibers[21,22]. The rapid diffusion could be aggravated in the system of conducting polymers due to the strong interactions of rigid chains. Thus, PAni molecules tend to bond into irregularly crystallized particles prior to undergoing extensive drawing, as present in the upper panels of Fig. 1a. In previous reports using poor solvents as coagulation bath, although CPFs with a smooth surface could be collected by enhanced shear flow and strong stretching[12,14], diameters are unable to be decreased to the ideal level due to the insufficient stretching slenderization of solidified gels. In contrast, the diffusion from dope fluids to good solvents is quite slow. Such slow diffusion allows the formation of fibrous gels with a low viscosity and the following high drawing ratios. Note that most conventional polymers are incapable of gelling in good solvents due to the poor chain interactions[23,24].

We calculated the diffusivities between various solvents and measured the viscosity of corresponding formed gels to support

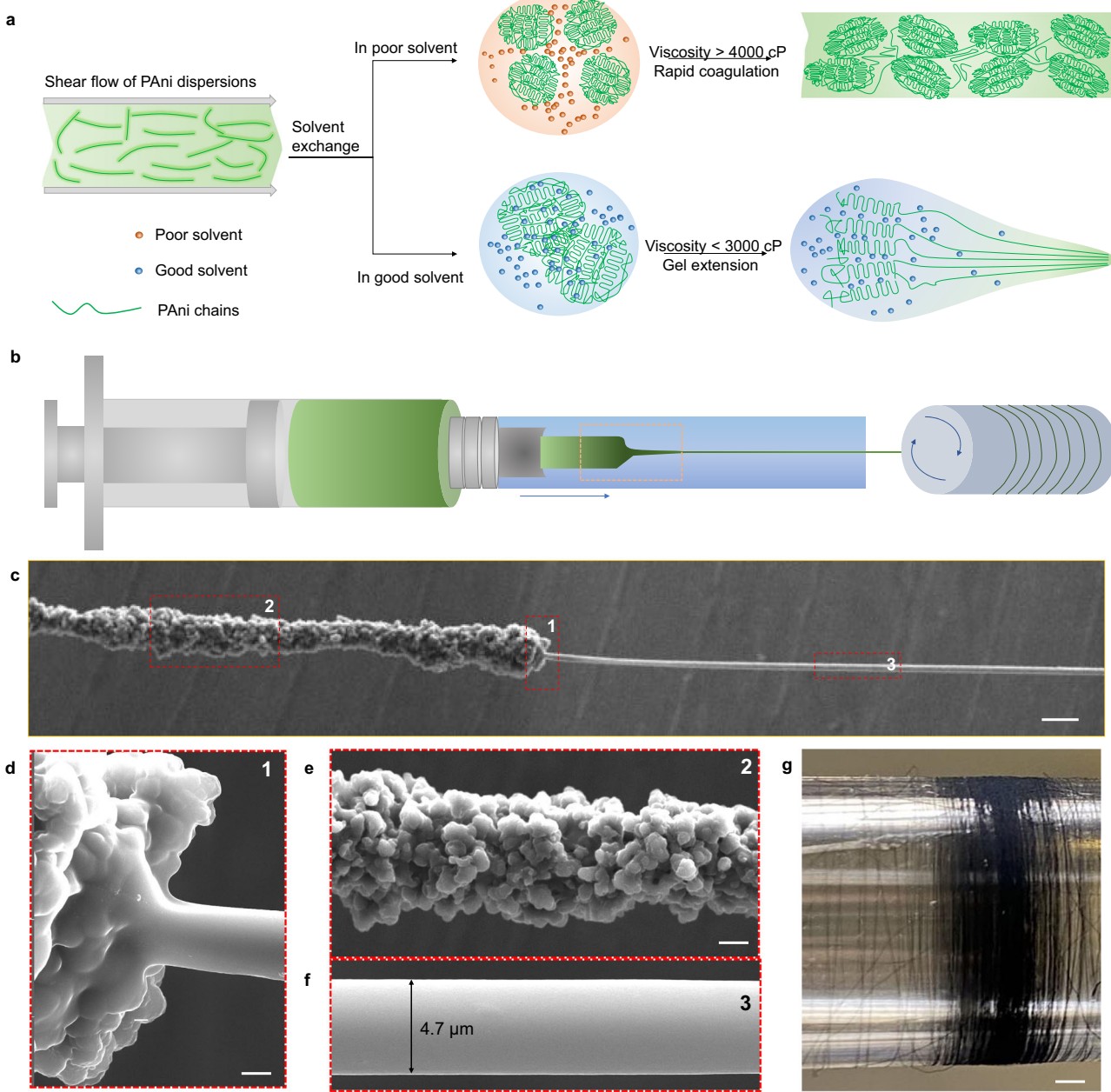

**Fig. 1 Scalable production of UFPFs. a** Schematic of the good solvent exchange strategy to prepare UFPFs in a modified wet spinning protocol. In the case of poor solvent exchange (light orange region, upper panel), PAni molecules are rapidly solidified into thick gels and protofibres with rough crystallized particles. In the case of good solvent exchange (light blue region, lower panel), the formed gels with low viscosity occur an impressive gel extension and are slenderized into ultrafine fibers. **b** Schematic of the modified wet spinning process. **c** Scanning electron microscope (SEM) image of the marked region in (**b**), showing the sharp necking behavior of gel PAni fibers. The close observation of region 1 (**d**), region 2 (**e**), and region 3 (**f**) in the marked zone of (**c**), illustrating the sharply necking process of PAni gels. **g** Photograph of a 5.4-kilometers-long UFPF collected in two hours. Scale bars: c 20 μm, d 2 μm, e 10 μm, g 150 mm.

our explanations. The diffusivity from A molecules to B molecules, $D_{AB}^0$ (cm$^2$ s$^{-1}$), is determined by Eq. (1):

$$\frac{D_{AB}^0 \mu_B}{T} = 8.52 \times 10^{-8} V_{bB}^{-1/3} \left[ 1.40 \left( \frac{V_{bB}}{V_{bA}} \right)^{1/3} + \frac{V_{bB}}{V_{bA}} \right] \quad (1)$$

where $\mu_B$ (cP) is the solvent viscosity, $T$ (K) is the temperature, and $V_b$ (cm$^3$ g$^{-1}$ mol$^{-1}$) is the molar volume of solvent at its normal boiling temperature[25]. As displayed in Fig. 2c, the diffusivities from m-cresol to DMF (7.5 × 10$^{-6}$ cm$^2$ s$^{-1}$) and NMP (7.71 × 10$^{-6}$ cm$^2$ s$^{-1}$) are generally lower than that of poor solvents. Diffusion in bath further dominates the viscosity of protofibres. To

monitor the viscosity of gel fibers in practical conditions, we conducted the viscometer tests at a low revolution (e.g., 10 Rev.). As summarized in Fig. 2d, the formed PAni gels in good solvents show a viscosity below 3000 cP, much lower than that of poor solvents (>4000 cP). The established solvating specie-diffusivity-viscosity formula accords well with our proposed explanations.

Interfacial pressure during the solvent exchange is another major factor relating to the necking behavior of PAni protofibres. In a two-fluid system, the interfacial pressure between two kinds of solvents is inclined to decrease with the improved solvent diffusion[26]. Based on the slow diffusion from m-cresol to good solvents (Fig. 2c), the interfacial pressure between gel fibers and

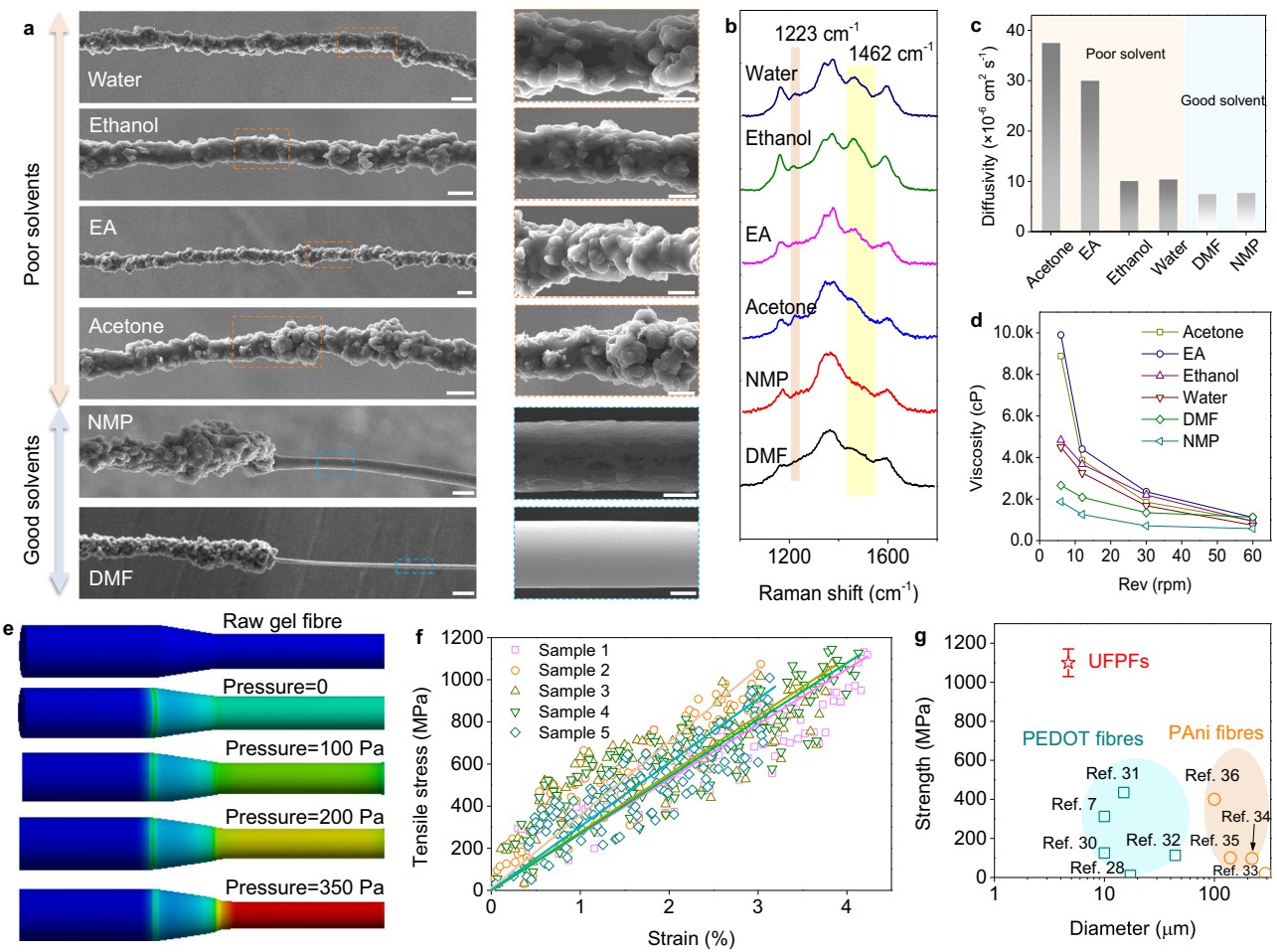

**Fig. 2 Mechanism and mechanical properties of UFPFs. a** SEM images of the PAni fibers produced in different solvating species. Specifically, the upper four panels show the fibers prepared from poor solvents, and the lower two panels show the fibers fabricated from good solvents. **b** Raman spectra of PAni fibers after placing in air for four weeks. **c** The diffusivity from PAni dispersions (in m-cresol) to various solvating species. **d** The viscosity of PAni gels formed in various solvating species. **e** Mechanics simulation results of extension behaviors of PAni gel fibers at different interfacial pressure. Comparing to the blue regions, the elements in blue regions are subject to larger stress. **f** Typical tensile stress-strain curves of UFPFs. **g** Ashby plot comparing the mechanical strength of UFPFs to previously reported CPFs. Scale bars in a: Water, Ethanol, EA, Acetone 20 μm (left) 10 μm (right); NMP 20 μm (left) 5 μm (right), DMF 20 μm (left) 2 μm (right).

the coagulation bath is considerable, which further induces the necking of protofibres. To understand this, we conducted a mechanic simulation of the stretching behavior of gel fibers at different interfacial pressures (see the progressive results in Supplementary Fig. 5 and "Method" section). According to the simulation results in Fig. 2e, the higher interfacial pressure drives gel fibers to occur the sharper necking and thinning effects at a given tensile stress. This probably explains the formation of UFPFs in DMF bath.

UFPFs show impressive mechanical performance. Different from that of conventional polymer fibers, the typical linear strain–stress curves of UFPFs demonstrate a brittle fracture behavior with a small tensile strain of 3.67 ± 0.64% (Fig. 2f). It is reasonable if considering the rigid backbone of PAni chains, which likely gather and condense into fragile fibrous assemblies after undergoing strong shear flow in spinning microtubes. According to classical Griffith theory on brittle fracture, fibers' strength generally improves with the decrease of diameter due to the depressed structural defects[27]. We compared the mechanical performance of UFPFs with previously reported CPFs.

Derived from the strain-stress curves, we concluded that UFPFs have a modulus of 29.89 ± 5.6 GPa, and strength of 1080 ± 71 MPa, at least one order of magnitude higher than that

of CPFs with larger diameters (Fig. 2g), mainly including PEDOT fibers (<450 MPa)[7,28–32] and PAni fibers (<400 MPa)[12,33–35].

**Energy and charge storage capacities**. Ultrafine morphology optimizes the electroactive surfaces, which enables UFPFs to exhibit superb energy and charge storage capacities. To evaluate the electrochemical activity of UFPFs, we constructed a micro capacitor using polyvinyl alcohol (PVA)-$H_3PO_4$ gel electrolyte and two UFPF electrodes (Fig. 3a). The electrochemical properties were checked by cyclic voltammetry (CV) and galvanostatic charge-discharge (GCD) measurements. At different scan rates, the nearly rectangular shape of CV curves and instantaneous current response to voltage reversal at each end potential suggest the good electrochemical activity of UFPFs[36] (Fig. 3b). The nearly triangular shape of GCD curves at different current densities illustrates the formation of efficient electric double layers and charge propagation across the UFPF electrodes[37] (Fig. 3c). According to the GCD results, we determined the electrochemical properties of UFPFs. Among them, the area capacitance, $C_A$, is between 1008 and 1666 mF cm$^{-2}$ at the current densities between 0.32 and 3.18 mA cm$^{-2}$, outperforming previously reported thick CPFs[27] and other electrodes, such as carbon nanomaterials[34,38],

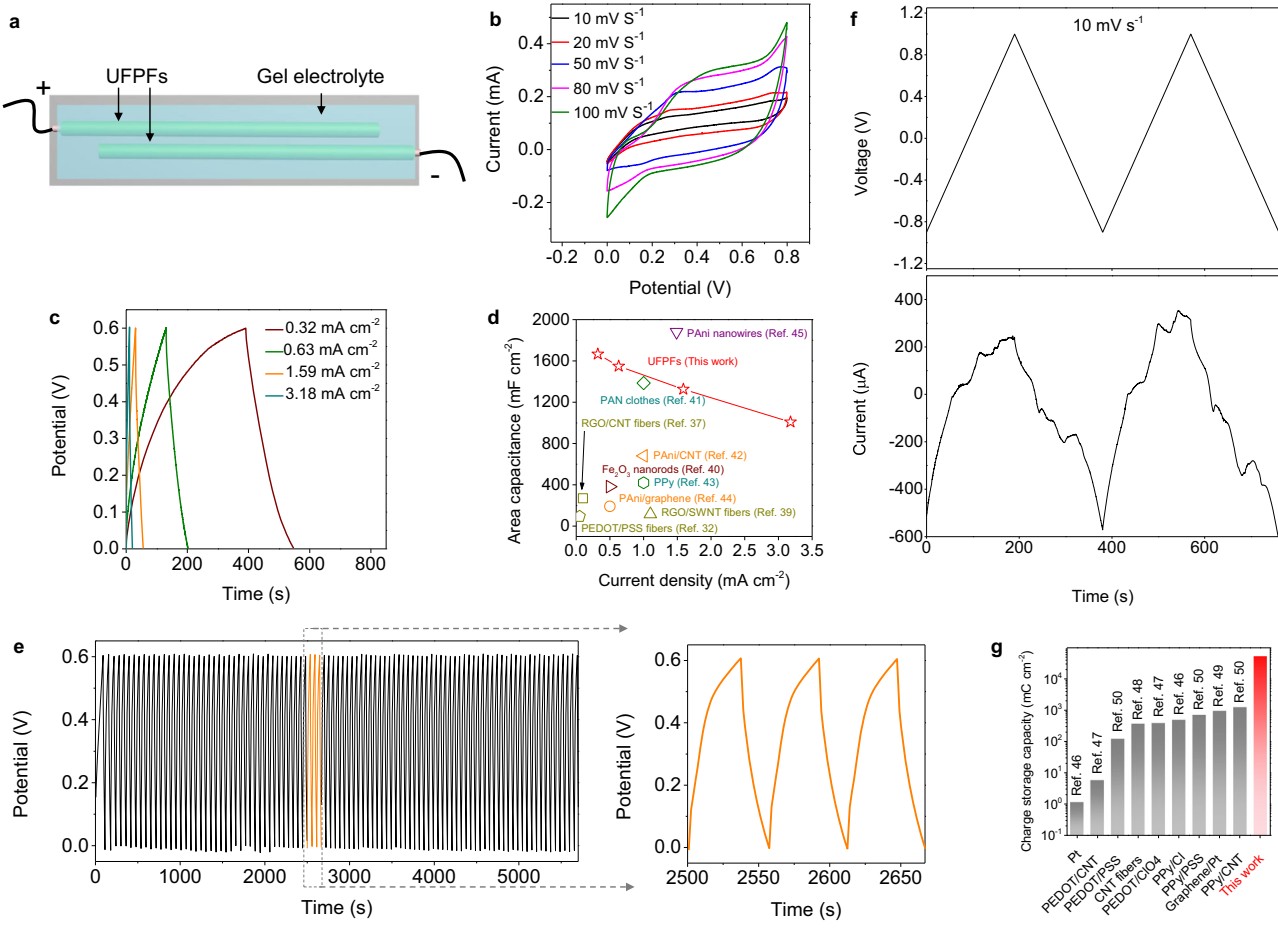

**Fig. 3 Energy and charge storage capacities of UFPFs. a** Schematic of a micro capacitor constructed using two UFPF electrodes on a substrate. **b** Cyclic voltammetry curves with the increasing scan rates from 10 to 20, 50, 80, and 100 mV s$^{-1}$. **c** Galvanostatic charge/discharge curves at various current densities increasing from 0.32 to 0.63, 1.59, and 3.18 mA cm$^{-2}$. **d** The area capacitance of UFPFs compared to previously reported electrodes. **e** Cycle galvanostatic charge/discharge curves during 120 cycles between 0 and 0.6 V at 1.59 mA cm$^{-2}$. **f** The relationship between current and voltage at a slow rate of 10 mV s$^{-1}$. **g** The charge storage capacity of UFPFs comparing to other charge storage materials.

metal oxides[39] and conducting polymers[40–43], and approaching to that of PAni nanowires[44] (Fig. 3d). The volumetric capacitance, power density, and energy density reached 83.8 F cm$^{-3}$, 0.96 W cm$^{-3}$, and 4.19 mWh cm$^{-3}$, respectively (Supplementary Fig. 6). In lifetime tests of UFPF-based capacitors, both the potential and capacitance continued without a significant decrease for 120 charge/discharge cycles at a low current density of 1.59 mA cm$^{-2}$, indicating the reliable electrochemical performance stability of UFPFs (Fig. 3e).

We were able to confirm the amount of transported charge per unit area to UFPF during the charge/discharge cycle. The charge during a triangular wave potential between −0.9 V and 1.0 V (water window, see Supplementary Fig. 7) was calculated by integrating the measured current with respect to the time of period at a low scan rate of 10 mV s$^{-1}$ (Fig. 3f)[6]. We determined that the charge storage capacity of UFPF was $5.25 \times 10^4$ mC cm$^{-2}$, a value at least two orders of magnitude higher than that of noble metals[45], carbon bulk[46–48] and previously reported conducting polymers[49] (Fig. 3g). This value decreases slightly to $2.015 \times 10^4$ mC cm$^{-2}$ at a tenfold scan rate of 100 mV s$^{-1}$ (Supplementary Fig. 8).

**Structure and performance of all-solid OECT.** Benefitting from the favorable energy and charge storage performance of UFPFs, we

demonstrated a high-performance all-solid OECT. OECT amplifies drain-source current intensities at low operating voltages by ion penetration into the organic mixed ionic-electronic conductors, i.e., conducting polymers[50,51]. This process is controlled by the gate bias, and, to date, has generally conducted in aqueous electrolytes. To preclude the interference of the external environment, we promoted the working conditions of OECT from aqueous environments to all-solid state by using gel electrolytes as the ion matrix. As shown in Fig. 4a, b, our OECT is mainly constructed by three polymer layers. The upper layer is the cured polyurethane (PU) working as the dielectric coating and protecting the device from the invasion of external action[52]. A fibrous silver gate electrode with a diameter of 7 µm is fixed in PU. Since UFPFs have demonstrated reliable electrochemical activities in PVA-H$_3$PO$_4$ gel, we used PVA-H$_3$PO$_4$ gel as the middle layer to inject ions to or uptake ions from the drain-source channel materials. A UFPF right below the silver gate is fused in the ion gel, and operates as the channel material. The bottom layer is also pure PU acting as the supporter of the whole device. Along the gate bias direction, the gate bias is mainly distributed in the gate/electrolyte interface and electrolyte/channel interface due to the almost insulating gel electrolyte (with a resistance beyond 10 MΩ). Due to the remarkable flexibility and transparency of PVA and PU, the all-solid OECT is very soft, and shows a transmittance beyond 80% in the region of visible light (Fig. 4c), and a small thickness below 300 µm.

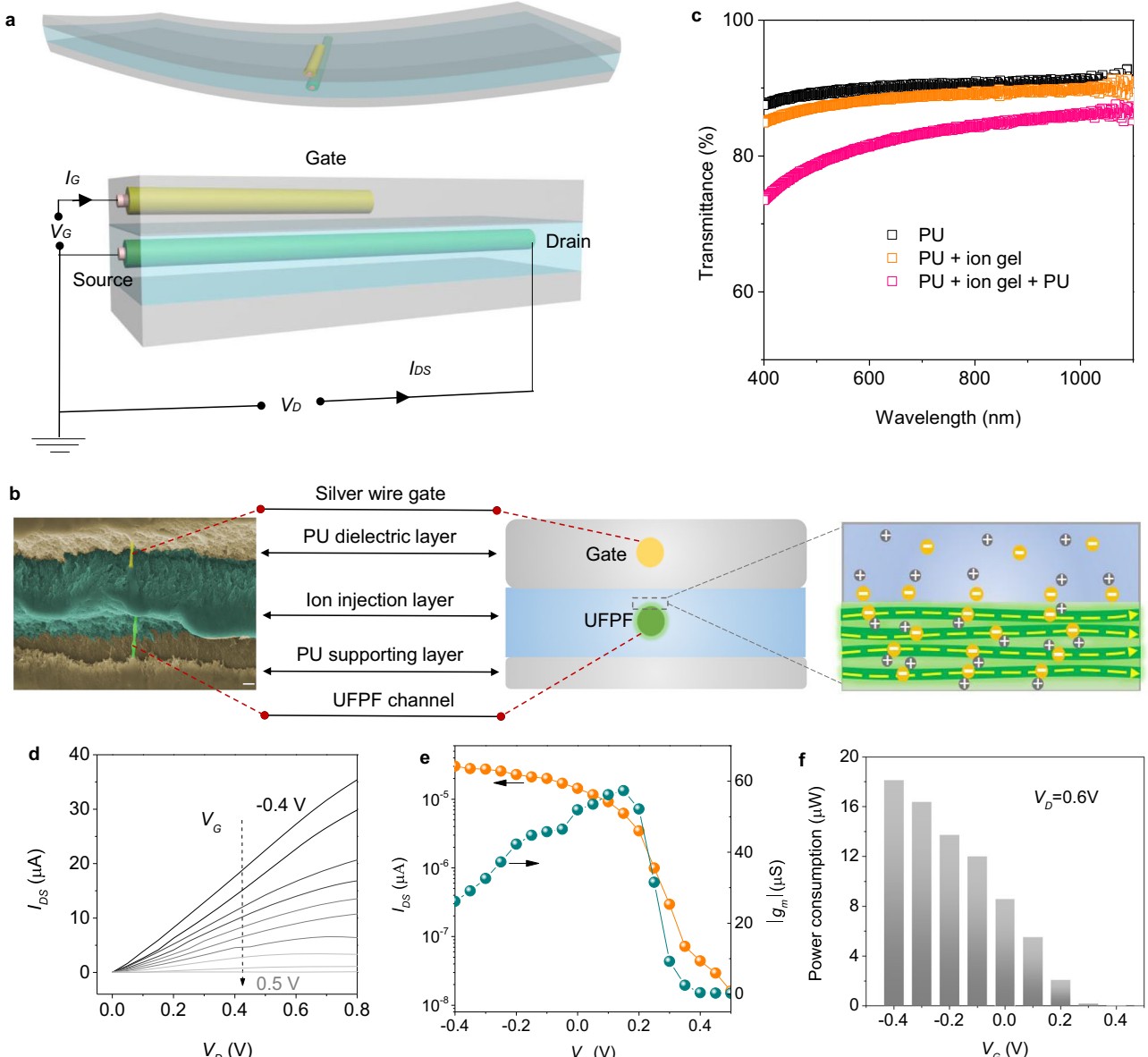

**Fig. 4 Demonstration and characterization of all-solid organic electrochemical transistor based on UFPFs. a** Schematic of the all-solid OECT composed of three polymer layers, one silver wire as the gate electrode, and one UFPF as the drain-source channel. **b** Cross-section SEM image and schematic of OECT. The yellow break lines direct the charge flow along the fiber chains (green solid lines). **c** Transmittance of the OECT in the region of visible light. A typical output curve (**d**), transfer curve (**e**), and power consumption in operation (**f**) of OECT. Scale bars: b 20 μm.

Despite the long channel length (~0.48 cm), much larger than that of a conventional micrometer-scale device, the all-solid OECT showed favorable amplification performance with a high on-off current ratio (>10³, Fig. 4e) at low voltages (<1 V, Fig. 4d). The relatively fair transconductance ($g_m$, < 60 μS) is probably ascribed to the small cross-sectional area, which dramatically magnifies the resistance of fibrillar channel. The comparison of electrical properties with previously reported fiber-based OECTs is listed in Supplementary Tab. 1. Note that the all-solid OECT is an energy-saving device with extremely low power consumption. For example, at a given drain-source voltage of 0.6 V, the consumed power is below 18 μW (Fig. 4f).

We proved that the all-solid OECT functioned to amplify small electrical signals in gel environments and respond to mechanical deformation as a tactile sensor. As illustrated in Fig. 5a, the applied vertical pressure on the surface of the all-solid OECT adjusted the ion penetration due to the improved gate-source

electric field and the redistribution of intrinsic capacitance[53]. At a $V_G$ of −0.1 V and a $V_D$ of 0.35 V, we observed a stable increase of drain-source current, $I_{DS}$, with the increasing pressure, up to a 92% amplification from 0 to 40 kPa (Fig. 5b). The sensitivity is at the level of 0.01–0.1 kPa$^{-1}$ in this process (dark cyan dots in Fig. 5b). As shown in Fig. 5c, the average rising time and falling time under instantaneous pressure of 17.8 kPa is ~536 ms and ~698 ms, respectively. Such integrated parameters facilitated the all-solid OECT to respond to different pressure levels from 0.92 to 22.2 kPa (Fig. 5d). In addition to the response to pressure in the vertical direction, the all-solid OECT also reacted to friction in the horizontal direction (Fig. 5e and Supplementary Fig. 9). The forward and backward friction of a load on the surface incurred horizontal movement of the PU layer, which drove the displacement of the Ag gate. Further, the real-time distance between the silver gate and the UFPF channel was changed repeatedly, thus producing a bimodal response curve (Fig. 5g).

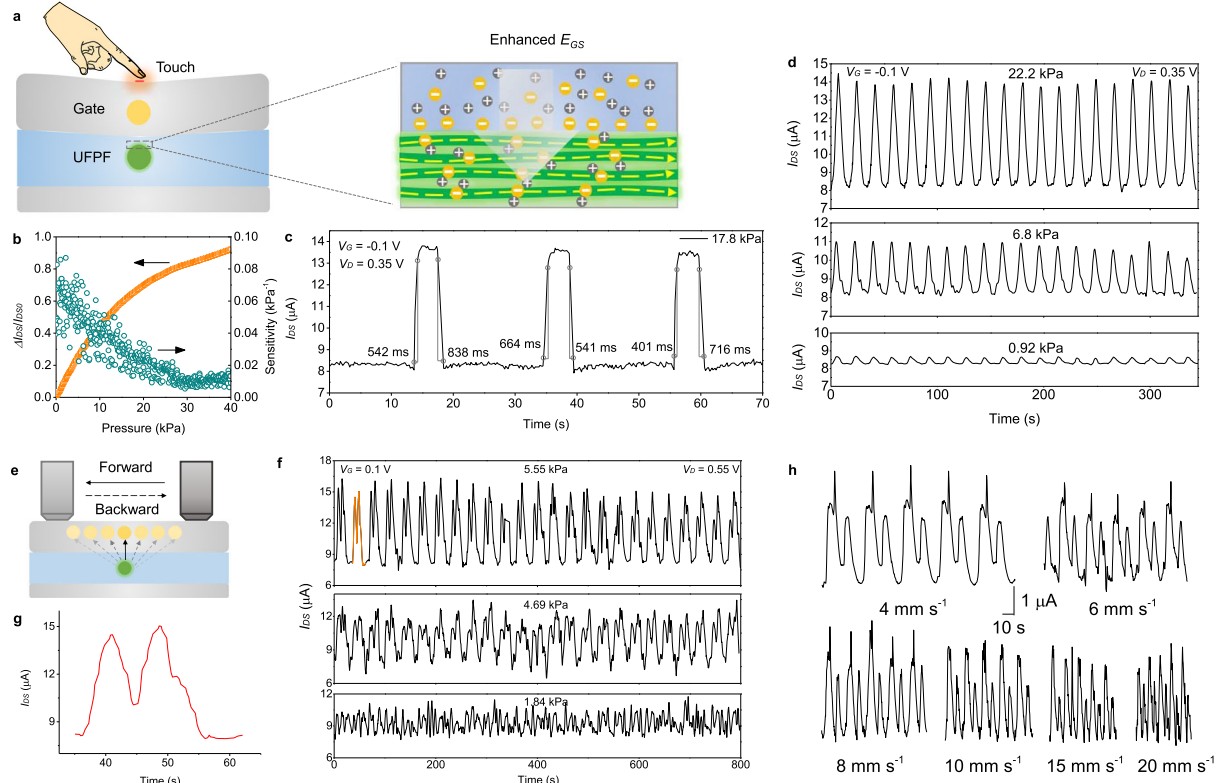

**Fig. 5 Electrical response of the all-solid OECT to mechanical deformations. a** Schematic of the mechanism explaining the response to the action of external pressure. **b** Relative drain-source change ($\Delta I_{DS}/I_{DS0}$) and sensitivity as a function of pressure. **c** Response time of the all-solid OECT when pressing (rising edge) and releasing (falling part) under the instantaneous pressure of 17.8 kPa. **d** Cyclic response at three different pressure levels (0.92, 6.8, and 22.2 kPa). **e**, Schematic of the working principle of the response to friction. **f** Cyclic response at three different frictions (1.84, 4.69, and 5.55 kPa). **g** An enlarged curve of the marked part in (**f**). **h** Cyclic response at different friction speeds from 4, 6, 8, 10, 15, to 20 mm s$^{-1}$.

Note that, to enable the enlargement of $I_{DS}$ with the increasing gate-channel distance under the repeated friction, we applied a positive $V_G$ of 0.1 V at a $V_D$ of 0.55 V. The all-solid OECT responded stably to friction at different magnitudes (Fig. 5f, from 1.84 to 5.55 kPa) and different speeds (Fig. 5h, from 4 to 20 mm s$^{-1}$) during our cyclic tests. For example, $I_{DS}$ increased ~86% at 5.55 kPa.

## Discussion

The past decades have witnessed great achievements in preparing high-performance CPFs, which made a vast difference to the rapid development of advanced electronics. However, due to the limitations of both technology and strategy, it is still difficult to produce ultrafine CPFs at a large scale. We proposed a good solvents strategy in modified wet spinning technology. With a principle of diffusion-controlled slow gelation of protofibres, the system successfully downsized the diameter of PAni fibers to below 5 μm, a value smaller than that of most previous work. Furthermore, the ultrafine morphology with highly improved electroactive surfaces promotes UFPFs to behave superb electrochemical activities and mechanical performance.

It is of great importance to realize the mass production of ultrafine CPFs. We constructed an all-solid OECT to employ the impressive energy and charge storage capacities of UFPFs. A handful of fibers are robust enough to satisfy the operation as the tactile sensor. In view of the ability to produce on the industrial scale, UFPFs are promised to be extended to large-area electronics, such as textile-scale numeric displays, soft electrochromic windows, and wearable energy harvesting systems.

## Methods

**Characterizations**. All the SEM images were collected on a tungsten thermionic emission SEM system (the Tescan VEGA3). XRD spectra were obtained from the XRD system (Rigaku SmartLab) equipped with a 9 kW rotating anode X-ray source (λ ~ 1.54 Å) coupling with a high-quality semiconductor detector that supports 0D, 1D, or 2D x-ray diffraction measurement. Raman spectra were recorded from Renishaw Micro-Raman Spectroscopy system fully integrated with a confocal microscope spectrometer and a 785 nm laser source. Mechanical tests were conducted on an advanced rheometric expansion system at the Hong Kong University of Science and Technology. All the electrochemical tests were processed on an electrochemical workstation (VersaSTAT3). The measurements of OECT were conducted on a probe station (Micromanipulator) with Keithley 4200A-SCS parameter analyzer. The test of viscosity was conducted by the viscometer (NDJ-5S/9 S/8 S). The probe of viscometer inserted into PAni gels after soaking in solvents, and the viscosity was measured at an increasing shear speed from 10 to 60 Rev.

**The fabrication of UFPFs**. PAni powder (emeraldine base, purchased from Aladdin, 25233-30-1) was mixed with CSA at a molar ratio of 2:1. After being milled for 15 min, the uniform doped PAni was dispersed in m-cresol (after degassing) at a concentration of 0.05 g mL$^{-1}$. The dispersions were used as spinning dopes after blending in air for 4–24 h, and extruded through a PEEK microtube with an inner diameter of 100 μm at a rate of 1 mL min$^{-1}$. A coagulation bath was chosen according to the experimental requirements. PAni fibers were directly drawn out from the bath and collected on a graphite roller continuously.

**Numerical method**. The experimental result is verified by a numerical method using commercial software ANSYS. The simulation is performed using workbench 18.0. In the simplified computational model, a geometric model of gel tube is developed, in which the ratio of diameter to length is chosen as 1:18, and the mechanical properties, density, Young's modulus, and Poisson's ratio are selected as 300 kg m$^{-3}$, 1000 Pa and 0.01, respectively. For the boundary conditions, one end of the gel model set as fixed support, and another end applies extended displacement to mimic the stretching effect in the actual situation. Meanwhile, the corresponding pressure is applied on the outer surface of the gel model to account

for the function of the impressive interfacial pressure on the surface of gel fibers. To ensure the convergence of the result, a grid independence test is conducted by refining mesh size sequentially, and the finite element mesh with 162641 nodes and 37128 hexahedral elements is adopted finally.

**The fabrication of micro capacitor.** Micro capacitor composed of two UFPF electrodes and the gel electrolyte was constructed on a glass substrate. To prepare the gel electrolyte, PVA power was dispersed into deionized water at a mass ratio of 9:1. PVA was dissolved after being heated for 5 hours at 85 °C. Then phosphoric acid was added at a mass ratio of 1:10 with deionized water. The mixture cooled at room temperature and was ready for use. Two UFPFs were placed in parallel on the glass slide. The transparent PVA-$H_3PO_4$ gel was dropped between UFPFs. Two copper wires connected to the UPPFs with silver paste worked as the conductor lines. After condensing for 10 min at 40 °C, the whole device was subject to electrochemical tests.

**The fabrication of all-solid OECT.** The OECT was built from three layers: two PU layers and one ion gel layer. PU dispersion in DMF was cast on a PVDF substrate. After being treated in the oven at 60 °C, a thin and transparent layer of pure PU was obtained. One drop of PVA-$H_3PO_4$ gel electrolyte was added to the surface of solidified PU. A UFPF was immersed in gel. After being dried at 45 °C for 15 min, a UFPF channel locked in PVA-$H_3PO_4$ gel was obtained. Afterward, another drop of PU was added and a silver wire operation as the gate electrode was put in PU at the liquid state. After being dried at 60 °C, an all-solid OECT was prepared. Note that all three electrodes were connected to cooper electrodes for the following measurements.

## Data availability

The authors declare that the data supporting the findings of this study are available from the corresponding author upon reasonable request.

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

## Acknowledgements

The authors are grateful for the financial support of the Research Grants Council of Hong Kong (No. 15201419), Hong Kong Polytechnic University Postdoctoral Fellowship and Endowed Professorship Fund (No. 847A). P.G. and Q.G. acknowledge funding from Shenzhen-Hong Kong Innovation Circle (No. SZSTI20EG14).

## Author contributions

X.T. supervised this study. B.F. designed and conducted the main experiments. J.Y., Y.C., and B.F. constructed and characterized the transistor. D.C. helped to build the wet spinning equipment and discussed the results. J.P. did the mechanic simulations. K.M.M. helped to draw a part of schematics. Q.G. and P.G. helped to conduct the mechanical tests. B.F. and X.T. wrote the manuscript.

## Competing interests

The authors declare no competing interests.
