## [Peer Review File · Nature Communications]

Scalable production of ultrafine polyaniline fibres for tactile organic electrochemical transistorsREVIEWER COMMENTS

Reviewer #1 (Remarks to the Author):

The research work reported in the paper represents a significant breakthrough in flexible, polymer-based electronic applications, particularly, (1) improved electrical performance, (2) a pathway for high-volume production, and (3) high mechanical performance. The fibers developed could be used for many other applications for flexible, wearable electronics beyond the examples highlighted in the paper. The production approach shows the promise for future commercial applications. The mechanical properties (strength, modulus) of the fibers are in the very high range, demonstrating the potential for many demanding applications. The paper is very well written, with a clear presentation of the state-of-the-art status, motivations, approaches, and mechanisms. The conclusions are reasonable and supported by data presented.

Comment regarding equation for D_{AB} : What is its source? From literature or by data fitting? Variables in equation are defined but units should also be given.

Minor edits: less one => less than one; larger than of conventional => larger than that of conventional

Reviewer #2 (Remarks to the Author):

Summary:

The authors report a method to develop PANI fibers using a wet spinning approach with small diameters (< 5 μm), good mechanical strength, and electrical properties amenable for use in e-textile applications. The reported method leveraged predoping with camphor sulfonic acid, and careful solvent choice of the coagulation bath. The authors demonstrate that choice of solvent dramatically impacts the surface morphology of the fibers, with NMP and DMF providing smooth and thin fibers with stability in air for four weeks. The authors then speculate that the use of a “good” solvent prevented quick solidification, and increased interfacial pressure during solvent exchange, and therefore allowed stretching of the fiber to reduce diameter and enhance morphology. The fibers showed good mechanical performance compared to other non-blended conjugated polymer-based fibers, which was attributed to the decrease in diameter and structural defects.

The fibers were then used to create a microcapacitor, which was characterized by CV and GCD. The capacitance, power density, energy density, and stability, demonstrated reliable electrochemical performance. The charge storage capacity was determined and rivals that of other similar platforms. As

such the fiber was used to make a solid-state OECT with a silver wire as the gate electrode, the fiber as the channel, and a PVA-H₃PO₄ gel, all wrapped in a polyurethane protective coating. The OECT was characterized and demonstrated low power consumption. The OECT was then demonstrated to respond to mechanical deformation through the pressure induced change in ion penetration. The favorable mechanical and electronic properties, as well as the scalability of these fibers will be of interest to the scientific community and warrants publication in Nature Communications. Please respond to the following:

- 1) What exactly do the authors mean by “good” or “bad” solvent? Solution viscosity? The speed at which the PANI solidifies/coagulates, preventing slenderization? PANI solubility? 2nd virial coefficient (A₂)? Diffusivity?
- 2) Why do the differences in morphology when using a “good” or “bad” solvent lead to differences in stability?
- 3) How reproducible and uniform are the properties of these fibers?
- 4) For the fiber based OECT, an insulating PU was used to encapsulate the Ag wire, how does the capacitance of the PU compare with the double layer capacitance near the fiber/solid-state electrolyte interface? How is the gate voltage distributing from the gate to the channel?
- 5) Can the authors make a table to compare the basic electrical properties of their OECT with previously reported fiber based OECTs

Figures

- When referring to the 5.4 km of UFPF, the text says Fig. 1e but should refer to Fig. 1g.
- Clearly indicate that the panel next to Fig. 3e is a zoom in.
- Add a legend to Fig. 2e to indicate the meaning of color
- For Fig. 5e, considering the relative position between the Ag wire and PU layer, why will a change in friction alter the position of Ag inside the PU?

Methods

- What molecular weight of PANI powder was used?
- Was there a particular choice behind using camphor sulfonic acid? Others have used AMPSA, sulfuric acid, HClO₄, and NMP.
- When referring to Raman “de-doping” peaks, explain what bonds you are looking at and give a citation.
- How did you perform your Rheology?

Point by point responses to the reviewer comments:

Reviewer 1:

1. **Reviewer's Comment:** The research work reported in the paper represents a significant breakthrough in flexible, polymer-based electronic applications, particularly, (1) improved electrical performance, (2) a pathway for high-volume production, and (3) high mechanical performance. The fibers developed could be used for many other applications for flexible, wearable electronics beyond the examples highlighted in the paper. The production approach shows the promise for future commercial applications. The mechanical properties (strength, modulus) of the fibers are in the very high range, demonstrating the potential for many demanding applications. The paper is very well written, with a clear presentation of the state-of-the-art status, motivations, approaches, and mechanisms. The conclusions are reasonable and supported by data presented.

Our response: We thank the reviewer for the positive comments.

2. **Reviewer's Comment regarding equation for D_{AB}^0 :** What is its source? From literature or by data fitting? Variables in equation are defined but units should also be given.

Our response: We thank the reviewer for kind concern. The equation for D_{AB}^0 is derived from Ref. 25 (*Smart Structures and Materials 2001: Electroactive Polymer Actuators and Devices. 4329, 59-71, 2001*). We have also added the units of every variable (highlighted in **Line 174-177**).

3. **Reviewer's Comment:** Minor edits: less one => less than one; larger than of conventional => larger than that of conventional.

Our response: We thank the reviewer for critically reading our manuscript and pointing out our mistakes. We have corrected all the edits (highlighted in **Line 34, 81 and 282**).

Reviewer 2:

1. **Reviewer's Comment:** The authors report a method to develop PANI fibers using a wet spinning approach with small diameters (< 5 μm), good mechanical strength, and electrical

properties amenable for use in e-textile applications. The reported method leveraged pre-doping with camphor sulfonic acid, and careful solvent choice of the coagulation bath. The authors demonstrate that choice of solvent dramatically impacts the surface morphology of the fibers, with NMP and DMF providing smooth and thin fibers with stability in air for four weeks. The authors then speculate that the use of a “good” solvent prevented quick solidification, and increased interfacial pressure during solvent exchange, and therefore allowed stretching of the fiber to reduce diameter and enhance morphology. The fibers showed good mechanical performance compared to other non-blended conjugated polymer-based fibers, which was attributed to the decrease in diameter and structural defects.

The fibers were then used to create a micro capacitor, which was characterized by CV and GCD. The capacitance, power density, energy density, and stability, demonstrated reliable electrochemical performance. The charge storage capacity was determined and rivals that of other similar platforms. As such the fiber was used to make a solid-state OECT with a silver wire as the gate electrode, the fiber as the channel, and a PVA-H₃PO₄ gel, all wrapped in a polyurethane protective coating. The OECT was characterized and demonstrated low power consumption. The OECT was then demonstrated to respond to mechanical deformation through the pressure induced change in ion penetration. The favorable mechanical and electronic properties, as well as the scalability of these fibers will be of interest to the scientific community and warrants publication in Nature Communications.

Our response: We thank the reviewer for the positive comments.

2. Reviewer's Comment: What exactly do the authors mean by “good” or “bad” solvent? Solution viscosity? The speed at which the PANI solidifies/coagulates, preventing slenderization? PANI solubility? 2nd virial coefficient (A₂)? Diffusivity?

Our response: We thank the reviewer for the important comment. From the perspective of physical interpretation, “good” and “poor” solvents of PANi are determined by the intermolecular interactions between PANi chains and solvent molecules. In a good solvent, interactions between PANi chains solvent molecules are energetically favorable, and will

cause PANi chains to expand and disperse well. In a poor solvent, PANi-PANi interactions are preferred and cause the PANi chains to disperse poor. Consequently, PANi molecules disperse good in good solvents, and disperse poor in poor solvents. We have added the discussions in the revised manuscript (highlighted in **Line 137-140**).

3. Reviewer's Comment: Why do the differences in morphology when using a “good” or “bad” solvent lead to differences in stability?

Our response: We thank the reviewer for the crucial questions. The morphology does not directly determine the differences in stability. According to our SEM observations and X-ray diffraction analysis, the fibres produced in good solvents behaved higher degree of orientation and crystallization. The higher degree of crystallization protects the doping bonding in PANi chains from the attack of ambientes at the molecular level. As a result, the fibres produced in good solvents could show better stability in air. To avoid the misleading, we have rearranged the related discussions in the revised manuscript (highlighted in **Line 145-150**).

4. Reviewer's Comment: How reproducible and uniform are the properties of these fibers?

Our response: We thank the reviewer for the kind comment. We discuss the uniformity of PANi fibres from the perspective of electrical and mechanical properties. In the case of electrical properties, because we used pre-doped PANi solutions as the raw dopes, the PANi fibres show uniform charge distribution throughout the fibre length. To confirm this point, we added a measurement to record the relationship between length and resistance. As shown in **Supplementary Fig. 4**, the value of resistance increases linearly with increasing length from 1 cm to 10 cm, demonstrating the favorable uniformity of electrical properties. In the case of mechanical properties, we provided the mechanical tensile tests of 5 fibres from 5 different batches, the results are shown in **Response Tab. 1** for the information. The tensile strength and strain are generally stable, demonstrating the reliable uniformity of mechanical properties. We have added the discussions in the revised manuscript (highlighted in **Line 111**).

Supplementary Fig. 4 The relationship between length and resistance of PAni fibres with increasing lengths from 1 cm to 10 cm.

Response Tab. 1: The mechanical tests of PAni fibre from 5 batches		
Batches	Tensile strength (MPa)	Strain (%)
Batch 1	1117.8	4.23
Batch 2	1073.2	3.03
Batch 3	1076.1	3.83
Batch 4	1127.8	4.13
Batch 5	1009.1	3.13

5. Reviewer's Comment: For the fiber based OECT, an insulating PU was used to encapsulate the Ag wire, how does the capacitance of the PU compare with the double layer capacitance near the fiber/solid-state electrolyte interface? How is the gate voltage distributing from the gate to the channel?

Our response: We thank the reviewer for the kind comments. We do not have the suitable devices at the micrometer scale to measure those parameters. However, PU is a dielectric layer which are difficult to be doped, so the capacitance of PU layer should be much smaller than that of fibre/electrolyte interface. From the gate to the channel, there are mainly three parts: the gate/electrolyte interface, gel electrolyte, and electrolyte/fibre interface. Among of them, the used gel electrolyte, PVA-H₃PO₄ gel, is almost insulating according to our

electrical test (with a resistance beyond 10 MΩ along with the gate bias direction). Thus, the gate bias is mainly divided into gate/electrolyte voltage and electrolyte/channel voltage. Kindly for your information, please refer to Ref. 54 (*IEEE Electron Device Lett.* 42, 46-49, 2020). We have added the discussions in the revised manuscript (highlighted in **Line 275-277**).

6. Reviewer's Comment: Can the authors make a table to compare the basic electrical properties of their OEET with previously reported fiber based OEETs?

Our response: We thank the reviewer for the kind suggestions. We have added the table to compare the electrical properties of our device to other fibre-based OEETs in the revised manuscript (highlighted in **Line 287-288**).

Supplementary Tab. 1: The electrical properties in fibre-based OEETs				
Ref No.	Channel	On/Off ratio	Drive (V)	g_m (μS)
This work	PAni	10^3	0.6	60
Ref. S1	PPy/PVA/PE	2.6×10^2	3	/
Ref. S2	PPy	10^4	2	/
Ref. S3	CNT	10^2	1	1350
Ref. S4	PPy/Graphene	10^2	2	/
Ref. S5	PEDOT/PSS	10^3	1	1000

7. Reviewer's Comment: When referring to the 5.4 km of UFPF, the text says Fig. 1e but should refer to Fig. 1g.

Our response: We have corrected it in the revised manuscript (highlighted in **Line 114**).

8. Reviewer's Comment: Clearly indicate that the panel next to Fig. 3e is a zoom in.

Our response: We thank the kind suggestions of the reviewer. We have added the label in **Fig. 3e** in the revised manuscript.

9. Reviewer's Comment: Add a legend to Fig. 2e to indicate the meaning of color.

Our response: We thank the reviewer for the kind suggestion. We have added the legend

in the revised manuscript (highlighted in **Line 130-132**).

10. Reviewer's Comment: For Fig. 5e, considering the relative position between the Ag wire and PU layer, why will a change in friction alter the position of Ag inside the PU?

Our response: We thank the reviewer for the kind comment. Our OECT is a polymer-based soft device. The Ag gate is located in the upper PU layer, close to the friction action interface. It feels like we rub our skins using fingers. The action of friction will incur repeated horizontal movement of PU layer, which drives the movement of Ag gate, as illustrated in **Fig. 5e**. We have added the discussions in the revised manuscript (highlighted in **Line 314-318**).

11. Reviewer's Comment: What molecular weight of PANI powder was used?

Our response: The molecular weight of PANi used is 91.1106. We are not sure the value of average chain length. It was purchased from Aladdin, and the CAS number is 25233-30-1. We have added the information of chemicals in the revised manuscript (highlighted in **Line 354**).

12. Reviewer's Comment: Was there a particular choice behind using camphor sulfonic acid? Others have used AMPSA, sulfuric acid, HClO₄, and NMP.

Our response: We thank the reviewer for the kind comment. The use of camphor sulfonic acid plays two roles: sulfonic groups enhance the dispersibility of PANi molecules and the proton improves the transport performance along PANi chains. Although we do not try other dopants due to the lack of storage in our lab, other dopants with above features could also work in principle.

13. Reviewer's Comment: When referring to Raman “de-doping” peaks, explain what bonds you are looking at and give a citation.

Our response: We thank the reviewer for the kind suggestion. We have added the explanations and some citations in the revised manuscript (highlighted in **Line 154-155**).

14. Reviewer's Comment: How did you perform your Rheology?

Our response: We thank the reviewer for the kind comment. The test of viscosity was conducted by the viscometer (NDJ-5S/9S/8S). We prepared a series of PANi gels by extruding PANi solutions in different solvents. Then, the probe of viscometer inserted into PANi gels after soaking in solvents, and the viscosity was recorded. To monitor the real state of gels as far as possible, we controlled the shear speed of probes at a very low value from 10 to 60 Rev. We have summarized and added the explanations in the revised manuscript (highlighted in **Line 351-353**).

REVIEWERS' COMMENTS

Reviewer #1 (Remarks to the Author):

The revised manuscript has addressed my concerns in my earlier review. I have no further comment.

Point by point responses to the reviewer comments:

Reviewer 1:

1. **Reviewer's Comment:** The revised manuscript has addressed my concerns in my earlier review. I have no further comment.

Our response: We thank the reviewer for the positive comments.